# Analysis of prognostic factors in critically ill patients with COVID-19

**Klaudia Bartoszewicz**[1], **Mateusz Bartoszewicz**[2], **Wojciech Gradkowski**[2], **Samuel Stróż**[1], **Anna Stasiak-Barmuta**[1], **Sławomir Lech Czaban**[2]*

**1** Department of Clinical Immunology, Medical University of Bialystok, Bialystok, Poland, **2** Department of Anaesthesiology and Intensive Care, Medical University of Bialystok, Bialystok, Poland

* slawomir.czaban@umb.edu.pl

**Data Availability Statement:** All relevant data are within the manuscript and its Supporting Information files.

## Abstract

The Coronavirus Disease 2019 (COVID-19) has caused a global health crisis. Mortality predictors in critically ill patients remain under investigation. A retrospective cohort study included 201 patients admitted to the intensive care unit (ICU) due to COVID-19. Data on demographic characteristics, laboratory findings, and mortality were collected. Logistic regression analysis was conducted with various independent variables, including demographic characteristics, clinical factors, and treatment methods. The study aimed to identify key risk factors associated with mortality in an ICU. In an investigation of 201 patients comprising non-survivors (n = 80, 40%) and Survivors (n = 121, 60%), we identified several markers significantly associated with ICU mortality. Lower Interleukin 6 and White Blood Cells levels at both 24- and 48-hours post-ICU admission emerged as significant indicators of survival. The study employed logistic regression analysis to evaluate risk factors for in-ICU mortality. Analysis results revealed that demographic and clinical factors, including gender, age, and comorbidities, were not significant predictors of in-ICU mortality. Ventilator-associated pneumonia was significantly higher in Survivors, and the use of antibiotics showed a significant association with increased mortality risk in the multivariate model (OR: 11.2, p = 0.031). Our study underscores the significance of monitoring Il-6 and WBC levels within 48 hours of ICU admission, potentially influencing COVID-19 patient outcomes. These insights may reshape therapeutic strategies and ICU protocols for critically ill patients.

## Introduction

The Coronavirus Disease 2019 (COVID-19), caused by the SARS-CoV-2 virus, emerged in late 2019 and rapidly evolved into a global pandemic, impacting healthcare systems and economies worldwide [1]. While most infected patients experience mild to moderate symptoms, a substantial fraction develop severe disease, demanding ICU admission and leading to significant mortality rates [2]. One of the most alarming aspects of the COVID-19 pandemic has been its impact on mortality, particularly among certain high-risk groups [3]. According to the World Health Organization (WHO), millions of confirmed cases and deaths have been

**Funding:** This study received partial funding from the Medical University of Bialystok. The funders had no role in study design, data collection and analysis, decision to publish, or preparation of the manuscript.

**Competing interests:** The authors declare no competing interests.

**Abbreviations:** BSI, Bloodstream Infection; VAP, Ventilator-Associated Pneumonia; MDR, Multi-Drug Resistant; MDRo, Multi-Drug Resistant Organisms; ICU, intensive care unit; DM, Diabetes Mellitus; AF, Atrial Fibrillation; HT, Hypertension; CHF, Chronic Heart Failure; APACHE II, Acute Physiology and Chronic Health Evaluation II; ICU, Intensive Care Unit; LOS, length of stay; MV, mechanical ventilation; APACHE II, Acute Physiology and Chronic Health Evaluation II; ARDS, Acute Respiratory Distress Syndrome; NMBAs, neuromuscular blocking agents; ECDC, European Centre for Disease and Control.

attributed to COVID-19 worldwide [4]. Mortality rates have varied by region, age group, and presence of underlying conditions, among other factors [5].

Several studies have been conducted to identify risk factors associated with COVID-19 mortality [3, 6]. Age, for instance, has been consistently identified as a strong predictor, with older individuals facing higher risks of severe outcomes [5]. Other factors like comorbidities, including diabetes, hypertension, and cardiovascular diseases, have also been shown to increase the risk of mortality [7]. Some studies have additionally focused on biomarkers, such as elevated D-dimer levels, C-reactive protein, and decreased lymphocyte counts as potential indicators of severe outcomes [8–10]. However, much of this research is generalized and does not specifically delve into predictors of mortality in critically ill patients, a gap this study aims to fill.

Severe manifestations of COVID-19 are typified by cytokine storm syndrome [11]. The cytokine storm is a critical and potentially lethal systemic inflammatory response. This phenomenon is characterized by the excessive and uncontrolled release of pro-inflammatory cytokines and chemokines by immune effector cells. In the context of COVID-19, the cytokine storm has gained particular attention due to its role in developing disease severity and complications [12]. Clinically, the cytokine storm could manifest as severe respiratory distress due to the development of ARDS, multi-organ failure, and coagulopathy, among other complications [12]. It is a major cause of morbidity and mortality in severe cases of COVID-19 and requires early recognition and treatment.

ARDS is a life-threatening form of respiratory failure. It is marked by the rapid onset of widespread inflammation in the lungs, alveolar damage, and severe hypoxemia. The pathophysiology of ARDS in COVID-19 is complex and multifaceted, involving direct viral-induced lung injury, dysregulated immune response, and endothelial damage [12].

Among the pro-inflammatory cytokines, IL-6 is commonly considered the most important pro-inflammatory cytokine in terms of its association with the pathogenesis of severe COVID-19 [11, 12]. Elevated levels of IL-6 have been consistently associated with disease severity and adverse outcomes [13].

Identifying prognostic factors is important to improve outcomes. With the ability to predict which patients are at higher mortality risk, healthcare providers can implement early and aggressive intervention strategies. Moreover, understanding these predictors can inform public health policies and contribute to developing clinical guidelines, thereby improving the standard of care for COVID-19 patients. The study aims to determine the mortality risk factors in critically ill patients with COVID-19.

## Materials and methods

### Study design and population

We conducted a retrospective cohort study at a single medical centre, adhering to STROBE guidelines. From March 3, 2020, to July 1, 2021, 235 adult patients diagnosed with COVID-19 were treated in the Intensive Care Unit (ICU) at the University Clinical Hospital in Bialystok, Poland. Data were collected from the electronic medical records from September 1 to November 30, 2021. Other research results regarding this group of patients [14]. The research investigator retrieved all the patient's data from the database in an Excel sheet and revised, cleaned, and coded the data. The study included participants at least 18 years old with a verified acute COVID-19 infection, confirmed through reverse transcription-polymerase chain reaction tests of nasal and throat swabs or lower respiratory secretions and ICU admission for SARS-CoV-2 infection.

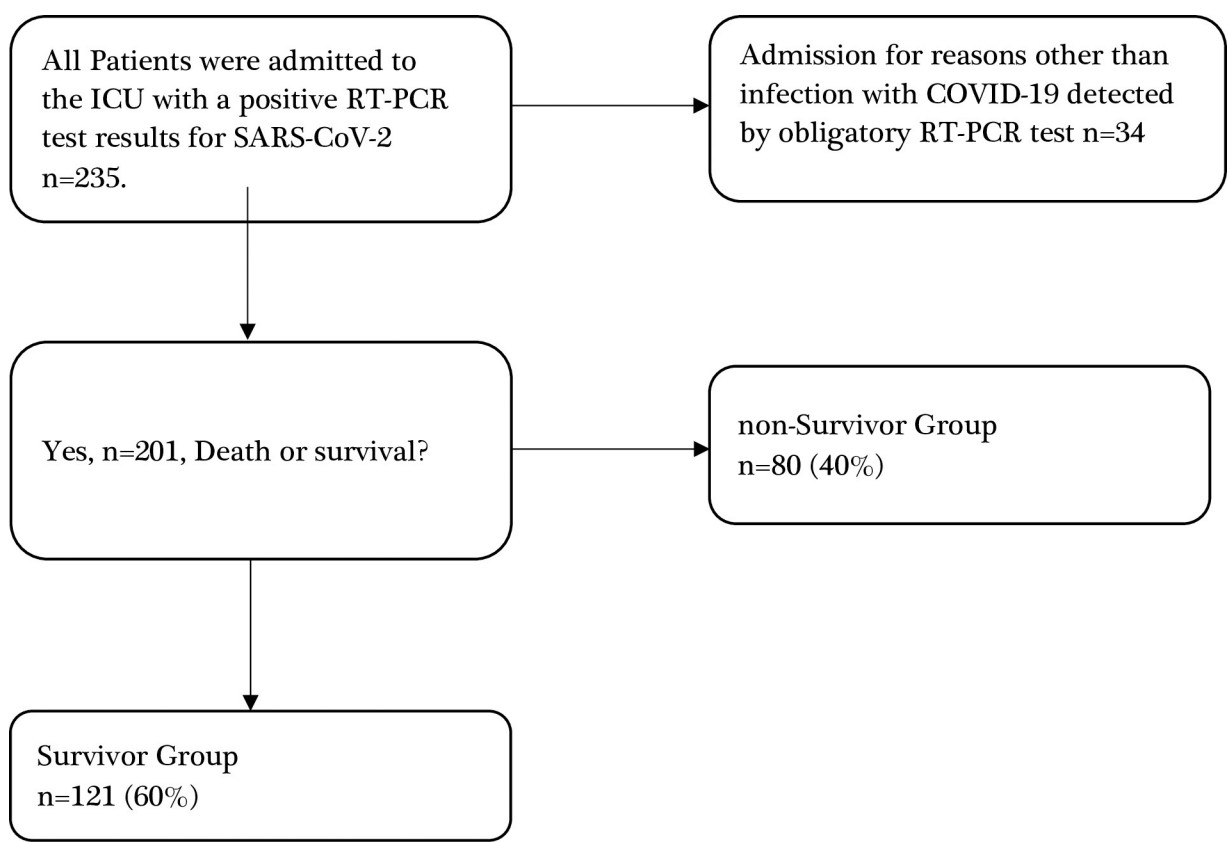

**Fig 1. Flowchart of patient screening and inclusion.** ICU: intensive care unit. SARS-CoV-2: severe acute respiratory syndrome coronavirus 2; COVID-19: coronavirus disease 2019; RT-PCR: reverse transcription-ploymerase chain reaction.

Excluded from this investigation were pregnant women and individuals in the ICU for non-COVID-19-related reasons, like elective surgeries or other emergency conditions. Consequently, the study comprised 201 participants, 80 (40%) of whom were categorized in the non-Survivor group. Details of the inclusion and exclusion criteria can be found in Fig 1's flowchart.

A single diagnostic laboratory at the Medical University of Bialystok, Poland, was used to avoid technical variables affecting laboratory results.

During the study period, the diagnostic laboratory of the Medical University of Bialystok employed two specific diagnostic tests: the Allplex 2019nCoV Assay (Seegene Inc., Seoul, South Korea) and the COVID-19 Real Time Multiplex RT-PCR Kit (Labsystems Diagnostics Oy, Vantaa, Finland). Both tests were cross-validated in the laboratory to ensure consistency and reproducibility of results. More information about the analysis of RT-PCR test results for SARS-CoV-2 diagnostics done in this paper is available in this research [15]. PCT was determined using the immunoassay method. IL-6 concentration was determined using the ELISA method. CRP concentration in blood serum was analyzed using the turbidimetric method. Morphology was analyzed using the automatic analyzer.

## Statistical analysis

Data from the patients were entered into a pre-established institutional database. A t-test or a chi-square test was employed to evaluate the relationship between various factors, reporting

the associated p-values. Logistic regression was also used to identify significant predictors affecting patient outcomes. A Logistic regression model included all variables in the univariate regression analysis with a p-value of less than 0.1. The aim was to determine factors independently predictive of illness progression. Two-tailed test, a p-value <0.05, was considered statistically significant. All statistical computations were conducted using R software version 4.1.1.

## Results

Table 1 presents the baseline and demographic characteristics of the 201 study participants, categorized into two groups: non-Survivors (n = 80, 40%) and Survivors (n = 121, 60%). No significant differences were observed between the Non-Survivors and Survivors in terms of average BMI (p = 0.474), sex distribution (p = 0.771), and mean age (p = 0.379). The prevalence of Diabetes Mellitus (DM), Atrial Fibrillation (AF), Hypertension (HT), and Chronic Heart Failure (CHF) were similar between the two groups (p>0.05 for all). Upon admission, APACHE II scores and Glasgow Coma Scale (GCS) were not significantly different between the two groups (p = 0.615 and p = 0.938, respectively). Inflammatory markers such as CRP, INR, and Interleukin 6 also showed no statistically significant differences (p>0.05). The length of stay (LOS) in ICU and the use of mechanical ventilation (MV) were similar in both groups (p = 0.551 and p = 0.450). Treatment modalities like the infusion of neuromuscular blocking agents (NMBAs) and corticosteroids did not vary significantly between the Non-Survivors and Survivors (p = 0.284 and p = 0.374, respectively). Ventilator-associated pneumonia (VAP) was significantly higher in the Survivor group (p = 0.048). Interleukin 6 levels after 48 hours of ICU hospitalization were significantly lower in the Survivor group (p<0.001). White Blood Cell counts after 24 and 48 hours of ICU hospitalization were significantly lower in the Survivor group (p = 0.012 and p = 0.022, respectively). Within 24 hours of ICU admission, non-survivors had significantly lower neutrophil percentages (60.3 ± 34.9 vs. 82.6 ± 19.4, p = 0.038), higher Interleukin-6 levels (448.4 ± 451.0 pg/mL vs. 206.1 ± 327.9 pg/mL, p = 0.050), and elevated white blood cell counts (15.9 ± 13.4 $\times 10^3$/μL vs. 11.4 ± 6.1 $\times 10^3$/μL, p = 0.012). These differences were sustained at 48 and 72 hours, most notably with Interleukin-6 levels being substantially higher in non-survivors at 48 hours (447.5 ± 390.8 pg/mL vs. 119.9 ± 154.5 pg/mL, p<0.001) and white blood cell counts consistently elevated across all time points (p = 0.022 at 48 hours, p = 0.034 at 72 hours).

Table 2 presents the results from the logistic regression analysis examining risk factors associated with in-ICU mortality. Univariate Analysis: Among demographic parameters, none of the assessed variables, including Gender, Age, BMI, and underlying medical conditions such as DM, AF, and CHF, were significantly associated with in-ICU mortality.

Only the neutrophil percentage was almost on the significance level in the univariate Analysis (OR: 1.014, 95% CI: 0.998–1.03, p = 0.078), similarly in the multivariate analysis (OR: 1.01, 95% CI: 0.998–1.03, p = 0.074). Ventilator-associated pneumonia (VAP) showed a significant association with in-ICU mortality (OR: 1.91, 95% CI: 1.021–3.56, p = 0.043), but this association did not remain significant in the multivariate analysis (OR: 1.0825, 95% CI: 0.43–2.7, p = 0.87). The use of antibiotics did not reach a level of statistical significance (OR: 2.252, 95% CI: 0.863–5.87, p = 0.097). However, in the multivariate analysis, the association became statistically significant (OR: 11.2, 95% CI: 1.2–101.66, p = 0.031).

## Discussion

Our study investigated various clinical and demographic variables to assess their potential impact on ICU mortality in 201 patients, non-Survivors (n = 80, 40%) and Survivors (n = 121, 60%). Traditional demographic and clinical parameters such as BMI, sex, age, and prevalent

**Table 1. Characteristics of patients with COVID-19 at ICU admission, risk factors for death, disease course, treatment.**

| Headcount | non-Survivor n = 80 (40%) | Survivor n = 121 (60%) | All n = 201 | *p*-value |
|---|---|---|---|---|
| **Baseline and demographic** | | | | |
| Average BMI (± SD) | 31.9 (7.4) | 33.7 (21.1) | 33.0 (17.0) | 0.474 |
| Female—no. (%) | 36 (45.0) | 51 (42.1) | 87 (43.3) | 0.771 |
| Mean age (± SD)—years | 67.0 (11.6) | 65.5 (12.4) | 66.1 (12.1) | 0.379 |
| DM | 25 (31.2) | 35 (28.9) | 60 (29.9) | 0.754 |
| AF | 11 (13.8) | 17 (14.0) | 28 (13.9) | 1.000 |
| HT | 50 (62.5) | 71 (59.2) | 121 (60.5) | 0.660 |
| Obesity | 17 (21.5) | 26 (21.7) | 43 (21.6) | 1.000 |
| CHF | 18 (22.8) | 28 (23.1) | 46 (23.0) | 1.000 |
| **On arrival in the ICU** | | | | |
| Mean APACHE II (± SD) | 29 (8) | 29 (8) | 29 (8) | 0.615 |
| Mean GCS (+SD) | 6.8 (5.4) | 6.7 (5.3) | 6.8 (5.3) | 0.938 |
| CRP (± SD) mg/L | 81.1 (88.5) | 85.5 (91.6) | 83.7 (90.2) | 0.735 |
| INR (± SD) | 1.3 (0.2) | 1.4 (0.3) | 1.4 (0.3) | 0.739 |
| Interleukin 6 (± SD) pg/mL | 580.9 (1031.3) | 372.5 (763.9) | 451.4 (876.9) | 0.186 |
| White Blood Cells ($\times 10^3$/µL) (±SD) | 13.6 (11.5) | 12.4 (8.0) | 12.9 (9.5) | 0.396 |
| Neutrophils Percent (± SD) | 69.2 (32.0) | 79.2 (21.2) | 75.2 (26.4) | 0.071 |
| Absolute neutrophils ($\times 10^3$/µL) | 11.2 (7.2) | 10.1 (6.7) | 10.5 (6.9) | 0.467 |
| Procalcitonin (± SD) ng/mL | 2.5 (9.8) | 2.1 (7.4) | 2.3 (8.4) | 0.780 |
| **During hospitalization** | | | | |
| Mean LOS at ICU (± SD)—days | 13.4 (10.2) | 12.5 (10.1) | 12.8 (10.1) | 0.551 |
| MV—no. (%) | 71 (88.8) | 112 (92.6) | 183 (91.0) | 0.450 |
| The average MV duration (± SD)—days | 11.2 (8.8) | 11.2 (9.1) | 11.2 (9.0) | 0.991 |
| Infusion of NMBAs at least one day (%) | 50 (62.5) | 85 (70.2) | 135 (67.2) | 0.284 |
| Corticosteroids—no. (%) | 68 (85.0) | 109 (90.1) | 177 (88.1) | 0.374 |
| Antibiotics—no. (%) | 69 (86.2) | 113 (93.4) | 182 (90.5) | 0.138 |
| Prone Position—no. (%) | 33 (41.2) | 46 (38.0) | 79 (39.3) | 0.661 |
| Bacterial coinfecion—no. (%) | 52 (65.0) | 89 (73.6) | 141 (70.1) | 0.211 |
| Fungal coinfecion—no. (%) | 20 (25.0) | 32 (26.4) | 52 (25.9) | 0.870 |
| MDR—no. (%) | 48 (60.0) | 83 (68.6) | 131 (65.2) | 0.457 |
| Sensitive—no. (%) | 4 (5.0) | 6 (5.0) | 10 (5.0) | 0.457 |
| VAP—no. (%) | 20 (25.0) | 47 (38.8) | 67 (33.3) | 0.048 |
| ARDS Mild | 6 (7.5) | 14 (11.6) | 20 (10.0) | 0.842 |
| ARDS Moderate | 31 (38.8) | 45 (37.2) | 76 (37.8) | |
| ARDS Not ARDS | 4 (5.0) | 5 (4.1) | 9 (4.5) | |
| ARDS Severe | 39 (48.8) | 57 (47.1) | 96 (47.8) | 0.842 |
| D-dimer (± SD) | 3.9 (4.6) | 3.9 (4.3) | 3.9 (4.4) | 0.985 |
| BSI—no. (%) | 15 (18.8) | 28 (23.1) | 43 (21.4) | 0.487 |
| Chronic organ insufficiency or immune compromise—no. (%) | 52 (65.0) | 76 (62.8) | 128 (63.7) | 0.767 |
| **After 24 hours of ICU hospitalization** | | | | |
| CRP (± SD) mg/L | 59.0 (72.1) | 76.2 (73.9) | 69.3 (73.4) | 0.177 |
| D-dimer (± SD) | 5.1 (5.7) | 5.2 (5.1) | 5.2 (5.3) | 0.928 |
| INR (± SD) | 1.3 (0.2) | 1.4 (0.3) | 1.4 (0.3) | 0.265 |
| Interleukin 6 (± SD) pg/mL | 448.4 (451.0) | 206.1 (327.9) | 297.0 (391.5) | 0.050 |
| Absolute neutrophils ($\times 10^3$/µL) | 10.9 (5.1) | 10.4 (5.6) | 10.6 (5.4) | 0.827 |
| Neutrophils Percent (± SD) | 60.3 (34.9) | 82.6 (19.4) | 74.6 (27.5) | 0.038 |
| Procalcitonin (± SD) ng/mL | 1.3 (2.9) | 2.2 (7.3) | 1.9 (6.0) | 0.409 |

*(Continued)*

**Table 1.** (Continued)

| Headcount | non-Survivor n = 80 (40%) | Survivor n = 121 (60%) | All n = 201 | *p*-value |
|---|---|---|---|---|
| White Blood Cells (×10³/μL) (±SD) | 15.9 (13.4) | 11.4 (6.1) | 13.2 (10.0) | 0.012 |
| **After 48 hours of ICU hospitalization** | | | | |
| CRP (± SD) mg/L | 47.8 (68.6) | 55.7 (65.5) | 52.6 (66.6) | 0.499 |
| D-dimer (± SD) | 4.3 (5.5) | 4.7 (4.6) | 4.6 (4.9) | 0.805 |
| INR (± SD) | 1.3 (0.2) | 1.3 (0.3) | 1.3 (0.2) | 0.166 |
| Interleukin 6 (± SD) pg/mL | 447.5 (390.8) | 119.9 (154.5) | 218.1 (287.6) | <0.001 |
| Absolute neutrophils (×10³/μL) | 11.7 (6.1) | 9.0 (5.8) | 9.8 (5.9) | 0.242 |
| Neutrophils Percent (± SD) | 64.9 (30.8) | 78.7 (17.2) | 74.5 (22.6) | 0.109 |
| Procalcitonin (± SD) ng/mL | 0.9 (1.5) | 1.7 (5.0) | 1.3 (4.0) | 0.272 |
| White Blood Cells (×10³/μL) (±SD) | 16.4 (14.1) | 12.0 (6.9) | 13.9 (10.8) | 0.022 |
| **After 72 hours of ICU hospitalization** | | | | |
| CRP (± SD) mg/L | 37.5 (46.3) | 52.5 (54.6) | 46.1 (51.6) | 0.109 |
| D-dimer (± SD) | 3.8 (3.9) | 4.7 (4.3) | 4.3 (4.1) | 0.474 |
| INR (± SD) | 1.3 (0.4) | 1.3 (0.2) | 1.3 (0.3) | 0.770 |
| Interleukin 6 (± SD) pg/mL | 476.8 (1084.6) | 736.6 (1233.5) | 621.8 (1163.8) | 0.474 |
| Absolute neutrophils (×10³/μL) | 12.7 (4.1) | 12.6 (6.4) | 12.6 (5.8) | 0.961 |
| Neutrophils Percent (± SD) | 58.8 (37.1) | 68.8 (27.8) | 66.2 (30.0) | 0.460 |
| Procalcitonin (± SD) ng/mL | 0.7 (1.2) | 1.0 (2.1) | 0.9 (1.8) | 0.353 |
| White Blood Cells (×10³/μL) (±SD) | 17.7 (14.1) | 13.3 (6.9) | 15.2 (10.7) | 0.034 |

The results are reported as a number (percentage) for categorical variables and median [IQR] and SD for continuous variables. DM: Diabetes Mellitus; AF: Atrial Fibrillation; HT: Hypertension; CHF: Chronic Heart Failure; APACHE II: Acute Physiology and Chronic Health Evaluation II; NMBAs: neuromuscular blocking agents; LOS: length of stay; ICU: Intensive Care Unit; VAP: ventilator-associated pneumonia, BSI: Bloodstream infection, 1 Median (IQR); n (%), 2 Wilcoxon rank sum test; Fisher's exact test; Pearson's Chi-squared test; Wilcoxon rank sum exact test, 3 False discovery rate correction for multiple testing

comorbidities showed no significant differences between the groups. We observed that VAP has a significant influence on in-ICU mortality. Interestingly, VAP occurrence was higher in Survivors. Initially, key inflammatory markers, including CRP and Il-6, did not differ significantly. However, over time, the most significant markers associated with survival were reduced levels of Il-6 after 24 and 48 hours and lower WBC counts after 24 and 48 hours of ICU admission.

The absence of significant differences in age, BMI, and pre-existing conditions indicate that these factors, usually considered crucial in determining outcomes, might be less predictive in ICU settings for COVID-19 patients, similar in research [6]. However, this is counter to other existing studies [16–18]. These results underscore the complexity of COVID-19's impact.

Findings highlight the role of the patient's immune response, particularly the inflammatory response, in influencing outcomes. The marked elevation of Il-6 in Non-Survivors is connected with cytokine storm, where an excessive immune response leads to tissue damage and organ failure. Furthermore, the higher WBC counts in Non-Survivors could indicate ongoing infection or inflammation, contributing to adverse outcomes. Lower Il-6 and WBC counts in survivors indicate a more controlled immune response, less likely to result in harmful inflammatory cascades [19]. It could be crucial for future targeted anti-inflammatory treatments for COVID-19 [11, 12, 20]. Medical teams could consider placing more emphasis on monitoring and controlling inflammatory markers within the first 48 hours of ICU admission. Il-6 and WBC can be used in more personalized treatment plans, moving away from a one-size-fits-all approach. More effective treatment protocols and targeted resource allocation based on predictive factors can reduce healthcare costs. Monitoring IL-6 could become a standard part of

**Table 2. Logistic regression of factors associated with in-ICU mortality.**

| Independent variables | Univariate Analysis | Multivariate Analysis |
|---|---|---|
| | OR (95%CI), p-value | OR (95%CI), p-value |
| Female | 1.12 (0.635–1.98, p = 0.690) | |
| Age | 0.989 (0.966–1.01, p = 0.378) | |
| BMI | 1.01 (0.984–1.03, p = 0.508) | |
| DM | 0.895 (0.484–1.66, p = 0.725) | |
| AF | 1.03 (0.453–2.32, p = 0.952) | |
| HT | 0.869 (0.486–1.55, p = 0.637) | |
| OBESITY | 1.01 (0.506–2.01, p = 0.980) | |
| CHF | 1.02 (0.520–2.00, p = 0.953) | |
| APACHE II | 0.991 (0.957–1.03, p = 0.613) | |
| White Blood Cells | 0.987 (0.959–1.02, p = 0.397) | |
| Interleukin 6 | 1.000 (0.999–1.00, p = 0.194) | |
| CRP | 1.00 (0.997–1.00, p = 0.734) | |
| Absolute neutrophils | 0.978 (0.921–1.04, p = 0.464) | |
| Neutrophils Percent | 1.014 (0.998–1.03, p = 0.078) | 1.01 (0.998–1.03, p = 0.074) |
| Procalcitonin | 0.995 (0.963–1.03, p = 0.779) | |
| LOS at ICU | 0.992 (0.965–1.02, p = 0.550) | |
| MV | 1.58 (0.598–4.16, p = 0.357) | |
| MV duration | 1.00 (0.969–1.03, p = 0.991) | |
| Prone Position | 0.874 (0.491–1.56, p = 0.646) | |
| Infusion of NMBAs at least one day | 1.42 (0.780–2.57, p = 0.253) | |
| Corticosteroids | 1.60 (0.681–3.77, p = 0.280) | |
| ANTIBIOTICS | 2.252 (0.863–5.87, p = 0.097) | 11.2 (1.2–101.66, p = 0.031) |
| VAP | 1.91 (1.021–3.56, p = 0.043) | 1.0825 (0.43–2.7, p = 0.87) |
| BSI | 1.30 (0.646–2.63, p = 0.458) | |
| MDR | 1.51 (0.814–2.81, p = 0.190) | |
| Bacterial coinfection | 1.50 (0.812–2.76, p = 0.196) | |
| Fungal coinfection | 1.08 (0.564–2.06, p = 0.819) | |

APACHE II Acute Physiology and Chronic Health Evaluation II, ICU Intensive Care Unit, LOS length of stay, MV mechanical ventilation, BMI Body Mass Index, NMBA neuromuscular blocking agent, VAP Ventilator-Associated Pneumonia, BSI Bloodstream Infection, DM: Diabetes Mellitus, AF: Atrial Fibrillation, HT: Hypertension, CHF: Chronic Heart Failure, OR: Odds Ratio.

ICU management, guiding interventions to modulate the immune response. Understanding the role of inflammatory markers like Il-6 may also have implications for other diseases characterized by systemic inflammation. Previous research has identified cytokine profiles, particularly elevated Il-6, as indicators of disease severity and poor prognosis in COVID-19 patients [13]. Our findings additionally indicate these markers directly with ICU mortality. Consequently, it highlights the potential of these markers as indicators for worsening conditions and prognosis in ICU settings.

Patients who survive long enough to develop certain conditions (coinfections) may differ fundamentally from those who die before these conditions occur. It could skew the results to make it appear as though these conditions are protective. Reverse Causality: The risk factor and mortality relationship is not unidirectional. For instance, the higher incidence of VAP in survivors infer that these patients had a longer period of mechanical ventilation, which increases the risk for VAP [21].

Our study has several limitations, including its retrospective and single-center design, which introduces inherent biases and limitations associated with the use of existing medical records. However, the large sample size and meticulous data collection strengthen the reliability of our conclusions.

## Conclusion

The study underscores the limited predictive value of traditional demographic and clinical factors for ICU mortality in COVID-19 patients, highlighting the critical role of dynamic immune response markers like Il-6 and WBC counts. These findings advocate for a more focused approach to early immune modulation in ICU management for COVID-19. It underscores the need for personalized treatment strategies and further research into the mechanisms of immune dysregulation in COVID-19.

## Supporting information

**S1 File.**
(DOCX)

**S1 Data.**
(XLSX)

## Author Contributions

**Conceptualization:** Klaudia Bartoszewicz, Sławomir Lech Czaban.

**Data curation:** Mateusz Bartoszewicz.

**Formal analysis:** Klaudia Bartoszewicz, Mateusz Bartoszewicz.

**Investigation:** Klaudia Bartoszewicz.

**Methodology:** Klaudia Bartoszewicz.

**Project administration:** Klaudia Bartoszewicz, Wojciech Gradkowski, Sławomir Lech Czaban.

**Resources:** Mateusz Bartoszewicz.

**Software:** Klaudia Bartoszewicz.

**Supervision:** Samuel Stróż, Anna Stasiak-Barmuta, Sławomir Lech Czaban.

**Validation:** Klaudia Bartoszewicz, Sławomir Lech Czaban.

**Visualization:** Mateusz Bartoszewicz, Wojciech Gradkowski.

**Writing – original draft:** Klaudia Bartoszewicz.

**Writing – review & editing:** Klaudia Bartoszewicz, Sławomir Lech Czaban.

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
