## [Decision Letter · Decision Letter 0]

3 Dec 2023

PONE-D-23-34297Analysis of prognostic factors in critically ill patients with COVID-19PLOS ONE

Dear Dr. Czaban,

Thank you for submitting your manuscript to PLOS ONE. After careful consideration, we feel that it has merit but does not fully meet PLOS ONE’s publication criteria as it currently stands. Therefore, we invite you to submit a revised version of the manuscript that addresses the points raised during the review process.

We look forward to receiving your revised manuscript.

Kind regards,

Benjamin M. Liu, MBBS, PhD, D(ABMM), MB(ASCP)

Academic Editor

PLOS ONE

Reviewers' comments:

Reviewer's Responses to Questions

**Comments to the Author**

1. Is the manuscript technically sound, and do the data support the conclusions?

Reviewer #1: Yes

Reviewer #2: No

2. Has the statistical analysis been performed appropriately and rigorously? 

Reviewer #1: Yes

Reviewer #2: No

3. Have the authors made all data underlying the findings in their manuscript fully available?

Reviewer #1: Yes

Reviewer #2: Yes

4. Is the manuscript presented in an intelligible fashion and written in standard English?

Reviewer #1: Yes

Reviewer #2: Yes

5. Review Comments to the Author

Reviewer #1: 1. There are some good review articles summarizing research status quo on risk factors etc IL-6. But this manuscript failed to cite these papers. The pathogenesis of SARS-CoV-2 in cytokine storm and ARDS as well as challenges of testing and interpretating these biomarkers should be discussed based on the following 2 references.

Role of Host Immune and Inflammatory Responses in COVID-19 Cases with Underlying Primary Immunodeficiency: A Review. J Interferon Cytokine Res. 2020 Dec;40(12):549-554. doi: 10.1089/jir.2020.0210. PMID: 33337932; PMCID: PMC7757688.

Clinical significance of measuring serum cytokine levels as inflammatory biomarkers in adult and pediatric COVID-19 cases: A review. Cytokine. 2021 Jun;142:155478. doi: 10.1016/j.cyto.2021.155478. Epub 2021 Feb 23. PMID: 33667962; PMCID: PMC7901304.

2. The authors should add more information on the methods of detecting SARS-CoV-2 and the biomarkers and other indexes under investigation.

3. Are statistical analysis two tailed or one tailed?

Reviewer #2: Summary of the research: The study aims to predict the mortality in critically ill COVID - 19 patients. The authors have meticulously collected data of 201 patients admitted in the ICU. It has been concluded that APACHE II scores , prone positioning and neuromuscular blocking agents are associated with ICU mortality among COVID-19 patients. The manuscript has been written in an intelligible fashion.

Major issues:

Issue 1: Statistical analysis

The protocol used to select the variables for uni-variate and multivariate regression analysis has not been clearly defined. The authors can perhaps refer to the reference 9 cited in the manuscript for improving the statistical analysis especially the regression analyses and also the analysis of the continuous variables.

1a. The authors have concluded that APACHE II score, emerged as significant predictor in multivariate analysis. But in table 1, the mean APACHE II score in the survivor and non survivor group and all the cases appear similar (28.8 (7.9%), 29.4 (8.2%), 29.1(8.0%)). So the conclusion regarding the APACHE II score being a predictor is not statistically sound.

1b. The authors have discussed that the duration of the mechanical ventilation (MV) is inversely related to the mortality risk. Here the duration is not specified. If we look at the table 1, the means in the survivor and non survivor group and all the cases again appear to be the same (11.2 (9.1%), 11.2 (8.8%) and 11.2 (9%)). So what does this conclusion mean?

Minor issue

There is an incomplete sentence where the reference 19 is cited.

The data may perhaps reveal better patterns if more sound regression analysis is conducted. Looking forward to the improved version of the data analysis and hence better conclusions in the manuscript.

6. PLOS authors have the option to publish the peer review history of their article (what does this mean?). If published, this will include your full peer review and any attached files.

Reviewer #1: No

Reviewer #2: **Yes: **Dr Prasanna N Bhat

---

## [Author Response · Author response to Decision Letter 0]

9 Feb 2024

Dear Reviewers,

I am responding to your comments and suggestions for our manuscript, "Analysis of prognostic factors in critically ill patients with COVID-19," with reference number ONE-D-23-34297. We sincerely appreciate your insightful feedback and have thoroughly revised our manuscript with your suggestions.

These revisions adequately address the concerns raised and significantly enhance the quality and clarity of our manuscript. We are grateful for the opportunity to improve our work based on your invaluable feedback.

Thank you for considering our revised submission. We look forward to your decision and remain hopeful for a favourable outcome.

Sincerely,

PhD Sławomir Lech Czaban

Department of Anaesthesiology and Intensive Care, Medical University of Bialystok, Poland

e-mail: slawomir.czaban@umb.edu.pl

phone number: +48604486369

---

## [Editor Report · Decision Letter 1]

1 Apr 2024

Analysis of prognostic factors in critically ill patients with COVID-19

PONE-D-23-34297R1

Dear Dr. Czaban,

We’re pleased to inform you that your manuscript has been judged scientifically suitable for publication and will be formally accepted for publication once it meets all outstanding technical requirements.

Kind regards,

Benjamin M. Liu, MBBS, PhD, D(ABMM), MB(ASCP)

Academic Editor

PLOS ONE
---

## [Editor Report · Acceptance letter]

18 Jun 2024

PONE-D-23-34297R1 

PLOS ONE

Dear Dr. Czaban, 

I'm pleased to inform you that your manuscript has been deemed suitable for publication in PLOS ONE. Congratulations! Your manuscript is now being handed over to our production team.

Kind regards, 

on behalf of

Dr. Benjamin M. Liu 

Academic Editor

PLOS ONE